

# *MonoPhy*: a simple R package to find and visualize monophyly issues

Orlando Schwery and Brian C. O'Meara

Department of Ecology and Evolutionary Biology, University of Tennessee, Knoxville, TN, USA

## ABSTRACT

**Background.** The monophyly of taxa is an important attribute of a phylogenetic tree. A lack of it may hint at shortcomings of either the tree or the current taxonomy, or can indicate cases of incomplete lineage sorting or horizontal gene transfer. Whichever is the reason, a lack of monophyly can misguide subsequent analyses. While monophyly is conceptually simple, it is manually tedious and time consuming to assess on modern phylogenies of hundreds to thousands of species.
**Results.** The R package *MonoPhy* allows assessment and exploration of monophyly of taxa in a phylogeny. It can assess the monophyly of genera using the phylogeny only, and with an additional input file any other desired higher order taxa or unranked groups can be checked as well.
**Conclusion.** Summary tables, easily subsettable results and several visualization options allow quick and convenient exploration of monophyly issues, thus making *MonoPhy* a valuable tool for any researcher working with phylogenies.

## INTRODUCTION

Phylogenetic trees are undoubtedly crucial for most research in ecology or evolutionary biology. Whether one is studying trait evolution (e.g., *Coddington, 1988*; *Donoghue, 1989*), diversification (e.g., *Gilinsky & Good, 1991*; *Hey, 1992*), phylogeography (*Avise et al., 1987*), or simply relatedness within a group (e.g., *Czelusniak et al., 1982*; *Shochat & Dessauer, 1981*; *Sibley & Ahlquist, 1981*), bifurcating trees representing hierarchically nested relationships are central to the analysis. Exactly because phylogenies are so fundamental to the inferences we make, we need tools that enable us to examine how reconstructed relationships compare with existing assumptions, particularly taxonomy. We have computational approaches to estimate confidence for parts of a phylogeny (*Felsenstein, 1985*; *Larget & Simon, 1999*) or measuring distance between two phylogenies (*Robinson, 1971*), but assessing agreement of a new phylogeny with existing taxonomy is often done manually. This does not scale to modern phylogenies of hundreds to thousands of taxa. Modern taxonomy seeks to name clades: an ancestor and all of its descendants (the descendants thus form a monophyletic group). Discrepancies between the new phylogenetic hypothesis and the current taxonomic classification may indicate that the phylogeny is wrong or poorly resolved. Alternatively, a well-supported phylogeny that conflicts with currently recognized groups might suggest that the taxonomy should be reformed. To identify such discrepancies, one can simply

Corresponding author
Orlando Schwery,
oschwery@vols.utk.edu

assess whether the established taxa are monophyletic. A lack of group monophyly however, can also be an indicator for conflict between gene trees and the species tree, which may be a result of incomplete lineage sorting or horizontal gene transfer. In any case, monophyly issues in a phylogeny suggest a potential error that can affect downstream analysis and inference. For example, it will mislead ancestral trait or area reconstruction or introduce false signals when assigning unsampled diversity for diversification analyses (e.g., in *diversitree* (*FitzJohn, 2012*) or BAMM (*Rabosky, 2014*)). In general, a lack of monophyly can blur patterns we might see in the data otherwise.

As this problem is by no means new, approaches to solve it have been developed earlier, particularly for large scale sequencing projects in bacteria and archaea, for which taxonomic issues are notoriously challenging. The program GRUNT (*Dalevi et al., 2007*) uses a tip to root walk approach to group, regroup, and name clades according to certain user defined criteria. The subsequently developed 'taxonomy to tree' approach (*McDonald et al., 2012*) matches existing taxonomic levels onto newly generated trees, allowing classification of unidentified sequences and proposal of changes to the taxonomic nomenclature based on tree topology. Finally, *Matsen & Gallagher (2012)* have developed algorithms that find mismatches between taxonomy and phylogeny using a convex subcoloring approach.

The new tool presented here, the R package *MonoPhy*, is a quick and user-friendly method for assessing monophyly of taxa in a given phylogeny. While the R package *ape* (*Paradis, Claude & Strimmer, 2004*) already contains the helpful function `is.monophyletic`, which also enables testing for monophyly, the functionality of *MonoPhy* is much broader. Apart from assessing monophyly for all groups and focal taxonomic levels in a tree at once, *MonoPhy* is also not limited to providing a simple 'yes-or-no' output, but rather enables the user to explore underlying causes of non-monophyly. In the following, we outline the structure and usage of the package and provide examples to demonstrate its functionality. For a more usage-focused and application-oriented treatment, one should refer to the tutorial vignette (`vignette` ("MonoPhyVignette")), which contains stepwise instructions for the different functions and their options. For any other package details, consult the documentation (`help`("MonoPhy")).

## DESCRIPTION

The package *MonoPhy* is written in R (*R Development Core Team, 2014*, http://www.R-project.org/), an increasingly important language for evolutionary biology. It builds on the existing packages *ape* (*Paradis, Claude & Strimmer, 2004*), *phytools* (*Revell, 2012*), *phangorn* (*Schliep, 2011*), *RColorBrewer* (*Neuwirth, 2014*) and *taxize* (*Chamberlain & Szocs, 2013*). A list of the currently implemented commands is given in Table 1. Note that in the code and this paper, we distinguish between tips, the organisms at the tip of the tree, and higher order taxa. Functions with 'taxa' only return information about higher order taxa, not tips. The main function—`AssessMonophyly`—evaluates the monophyly of each higher order taxon by identifying the most recent common ancestor (MRCA) of a collection of tips (e.g., all species in a genus), and then returning all descendants of this node. The taxon is monophyletic if the number of its members (tips) equals the number of

**Table 1**  Functions of the package *MonoPhy*.

| Function name | Description |
| --- | --- |
| AssessMonophyly | Runs the main analysis to assess monophyly of groups on a tree. |
| GetAncNodes | Returns MRCA nodes for taxa. |
| GetIntruderTaxa | Returns lists of taxa that cause monophyly issues for another taxon. |
| GetIntruderTips | Returns lists of tips that cause monophyly issues for a taxon. |
| GetOutlierTaxa | Returns lists of taxa that have monophyly issues due to outliers. |
| GetOutlierTips | Returns lists of tips that cause monophyly issues for their taxon by being outliers. |
| GetResultMonophyly | Returns an extended table of the results. |
| GetSummaryMonophyly | Returns a summary table of the results. |
| PlotMonophyly | Allows several visualizations of the result. |

descendants of its MRCA. If there are more descendants than taxon members, the function will identify and list the tips that do not belong to the focal taxon and we then call these tips 'intruders.' Accordingly, we will further refer to the taxa whose monophyly was disrupted by these 'intruders' as 'intruded.' Note that if two taxa are reciprocally disrupting each other's monophyly, certain tips of intruded taxa will often be intruders themselves: if the phylogeny is ((A1, B1), (A2, B2)), where A and B are genera, it is not clear if the A tips are intruding in B or the B tips are intruding in A.

Biologically, identifying a few intruders may suggest that the definition of a group should be expanded; observing some group members in very different parts of the tree than the rest of their taxon may instead suggest that these individuals were misidentified, that their placement is the result of contaminated sequences or due to horizontal gene transfer between members of two remote clades. Moreover, the approach as described above would suggest that the clades that are intruded by the outlier tips would in turn be intruders to the taxon the outliers belong to, which intuitively would not make sense. We thus implemented an option to specify a cutoff value, which defines the minimal proportion of tips among the descendants of a taxon's MRCA that are labeled as being actual members of that taxon. If a given group falls below this value, the function will find the 'core clade' (a subclade for which the proportion matches or exceeds the cutoff value) by moving tipward, always following the descendant node with the greater number of tips in the focal taxon (absolute, relative if tied), and at each step evaluating the subtree rooted at that node to see if it exceeds the cutoff value. Once such a subtree is found, it is then called the 'core clade', and taxon members outside this clade are then called 'outliers'. As there is no objective criterion to decide at what point individuals should be considered outliers, a reasonable cutoff value must be chosen by the user.

If the tree's tip labels are in the format '*Genus_speciesepithet*', the genus names will be extracted and used as taxon assignments for the tips. If the tip labels are in another format, or other taxonomic levels should be tested, taxon names can be assigned to the tips using an input file. To avoid having to manually compose a taxonomy file for a taxon-rich phylogeny, *MonoPhy* can automatically download desired taxonomic levels from ITIS or NCBI using *taxize* (*Chamberlain & Szocs, 2013*).

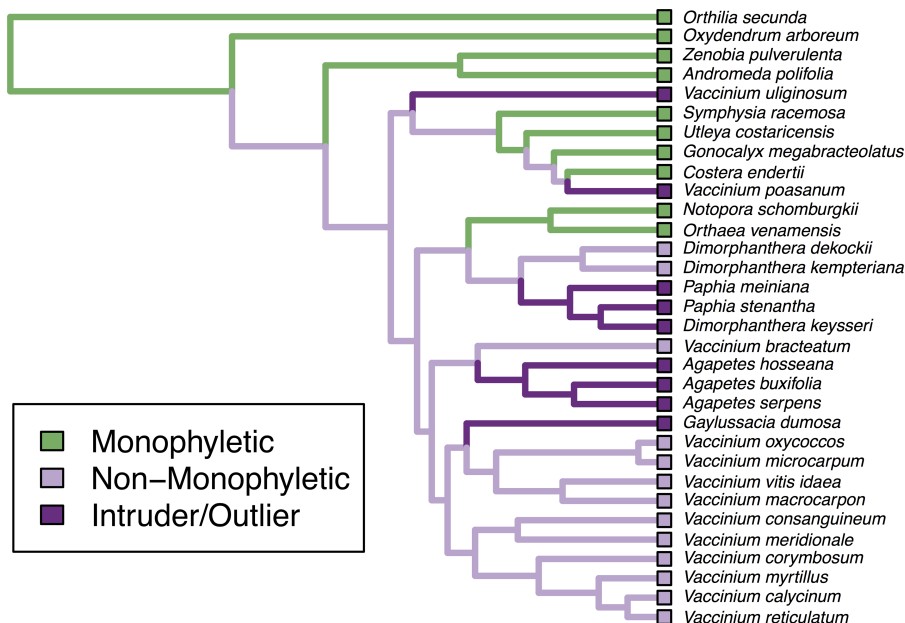

**Figure 1 Monophyly plot of the genera of Ericaceae.** Close-up on subfamily Vaccinioideae only. Branches of the tree coloured according to monophyly status. We can see that *Vaccinium* has two outliers and that its intruders are *Paphia*, *Dimorphanthera*, *Agapetes* and *Gaylussacia*.

All inference results are stored in a solution object, from which the other functions can extract information (e.g., summary tables, intruder and outlier lists) for one or more higher-level taxa of interest. `PlotMonophyly` reconstructs and plots the monophyly state of the tips using *phytools* (*Revell, 2012*). Apart from the basic monophyly plot (Fig. 1), branches can be coloured according to taxonomic groups or to highlight intruders and outliers. Monophyletic groups can be collapsed and plots can be saved directly to PDF to facilitate the visualization of large trees.

It is important to remember that the results produced by the package are merely the product of the used phylogeny and the available taxonomic information. It thus only makes the mismatches between those accessible, but does not reveal any more than that. The decision of whether the result suggests problems in the phylogeny or the taxonomy, whether a tip should be considered a rogue taxon and be removed or whether gene tree-species tree conflicts should be investigated, is entirely up to the user's judgment.

*MonoPhy* is available through CRAN (https://cran.r-project.org/package=MonoPhy/) and is developed on GitHub (https://github.com/oschwery/MonoPhy). Intended extensions and fixes can be seen in the issues list of the package's GitHub page. Among the planned extensions of the package are: multiple trees, displaying the result for specific subtrees, proposing monophyletic subgroups, enabling formal tests for monophyly (incorporating clade support) and providing increased plot customizability.

## EXAMPLES

Our first example makes use of the example files contained in the package. They come from a phylogeny of the plant family Ericaceae (*Schwery et al., 2015* pruned to 77 species; for

original data, see *Schwery et al., 2014*) and two taxon files assigning tribes and subfamilies to the tips (in both files, errors have been introduced for demonstration purposes; see code and output for both examples in Supplemental Information 1). Running the main analysis command `AssessMonophyly` on genus level (i.e., tree only) and tribe level (i.e., tree plus taxonomy file) using standard settings took 0.045 and 0.093 s respectively on a MacBook Pro with 2.4 GHz Intel Core i5 and 8 GB Ram. We could now use the remaining commands to extract the information of interest from the saved output object (e.g., summary tables, lists of problem taxa, etc.). The basic monophyly plot for the genus level analysis is displayed for a subclade of the tree in Fig. 1 (the figure of the full tree is shown in Fig. S1).

For the second example, we demonstrate the package's performance on a tree of 31,749 species of Embriophyta (*Zanne et al., 2014*; data see *Zanne et al., 2013*), using an outlier-cutoff of 0.9 this time. Just checking monophyly for genera took 1.78 h, but revealed that 22% of genera on the tree are not monophyletic, while around half of all genera are only represented by one species each. Furthermore, we can see that the largest monophyletic genus is *Iris* (139 tips), that *Justicia* had the most intruders (13 tips) and that *Acacia* produced the most outliers (99 tips). Finally, with 2,337 other tips as descendants of their MRCA, the 3 species of *Aldina* are most spread throughout the tree.

## CITATION

Researchers using *MonoPhy* in a published paper should cite this article and indicate the used version of the package. The citation information for the current package version can be obtained using `citation("MonoPhy")`.

## ACKNOWLEDGEMENTS

We want to thank the members of the O'Meara lab for helpful discussions, Frederik Matsen IV and two anonymous reviewers for well-considered criticism to improve the manuscript, Brian Looney and Sam Borstein for beta testing, and the members of the Tank lab, Arne Mooers, Karen Cranston, Bruce Cochrane and Daniel Gates for great ideas on increasing the usefulness of this package.

### Funding

This work has been supported via a GTA to OS by the University of Tennessee, Knoxville. The funders had no role in study design, data collection and analysis, decision to publish, or preparation of the manuscript.

### Grant Disclosures

The following grant information was disclosed by the authors:
University of Tennessee, Knoxville.

### Competing Interests

The authors declare there are no competing interests.

## Author Contributions

- Orlando Schwery conceived and designed the experiments, performed the experiments, analyzed the data, contributed materials/analysis tools, and wrote the paper, prepared figures and tables, performed the computation work.
- Brian C. O'Meara contributed materials/analysis tools, and wrote the paper.

## Data Availability

CRAN: https://cran.r-project.org/package=MonoPhy/

GitHub: https://github.com/oschwery/MonoPhy.

## Supplemental Information

Supplemental information for this article can be found online at http://dx.doi.org/10.7717/peerj-cs.56#supplemental-information.

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
