# Peer review of "MonoPhy*: a simple R package to find and visualize monophyly issues"

_PeerJ Computer Science, doi:10.7717/peerj-cs.56_

## Round 0.1 · original submission · Minor Revisions

As you can see the reviewers were generally positive about your article. All reviewers, however, raised several specific comments that should be addressed in a revised manuscript. These comments include a better discussion of monophyly and reasons why this property may not be satisfied, as well as small changes to the software to make it more useable.

Reviewer 1 ·

Basic reporting

The article is easy to understand and well written.

My major comment is that the authors state that higher order taxa not being monophyletic might be caused by problems in the taxonomy (i.e., mis-annotation) or problems with the data (i.e., contamination). One of the most important (but not discussed) reasons is gene tree/species tree conflict. This could be a result of incomplete lineage sorting (ILS) or hortizontal gene transfer (HGT). I think some discussion about this would help strengthen the paper. For example, an intruder species might be due to error (as suggested by the authors), or it might be a real biological finding due to HGT. The authors could then motivate MonoPhy as not only being able to identify error but also potentially HGT events.

Finally, there were a few minor corrections needed:

Page 2, Line 64: higher taxon -> higher order taxon
Page 3, Line 89: from by all -> from all

Experimental design

No comments

Validity of the findings

No comments

Additional comments

I have tested this software and it is easy to use. My only suggestion is that it would be helpful if the taxonomy file allowed header information. Thus, instead of using this command:

GetIntruderTips(solution1, taxa="Ericoideae", taxlevels=2)

One could use this command:

GetIntruderTips(solution1, taxa="Ericoideae", taxlevels='order')

As I have many different metadata files with header information beyond just taxonomy (i.e., geographic location, host organism, etc..) I could then easily write a script that could use any of my metadata files without needing to know exactly which column number is the group I'm interested in, I could just use the column name.

·

Basic reporting

In this paper, Schwery and O’Meara describe a new R package they have written, MonoPhy, for assessing taxonomic monophyly of phylogenetic trees. The package takes in phylogenetic trees with taxonomic leaf labels, and from that input makes plots and finds “intruder” taxa, which are taxa that disrupt monophyly. This should be a helpful package for researchers using R. The paper is written as a software announcement, and is generally suitable for that format.

However, I was left wanting clearer definitions of the objects under discussion. In particular, I was confused by the definitions of “intruders” and “outliers”. If I parse the definition of intruders correctly, “intruded” taxa will commonly themselves be intruders as well. For example, imagine that we have a large clade that is monophyletic with the exception of two somewhat widely spaced leaves. These two leaves will be marked as “intruders”, as should be the case, but by the definition the leaves below their MRCA will also be marked as intruders, which seems strange. Is this the case? I shouldn’t have to read the source to understand.

The defintion of outliers starts with the sentence “We thus implemented an option to specify a cutoff value, which gives the minimal proportion of tips among the descendants of a taxon’s MRCA that are actual members of that taxon.” I think this is supposed to mean “are assumed to be actual members of that taxon”? Again, it’s important to make these definitions clear.

Please describe the algorithm the code uses to find these various characteristics of the trees in the paper.

The paper does not cover relevant prior literature. In fact, I couldn’t find any reference to computational approaches to assessing concordance between a phylogeny and a taxonomy. Here are some relevant papers:

Dalevi, D., Desantis, T. Z., Fredslund, J., Andersen, G. L., Markowitz, V. M., & Hugenholtz, P. (2007). Automated group assignment in large phylogenetic trees using GRUNT: GRouping, Ungrouping, Naming Tool. BMC Bioinformatics, 8, 402.

This next paper (cited 667 times) builds on that one, developing the `tax2tree` tool:

McDonald, D., Price, M. N., Goodrich, J., Nawrocki, E. P., DeSantis, T. Z., Probst, A., … Hugenholtz, P. (2011). An improved Greengenes taxonomy with explicit ranks for ecological and evolutionary analyses of bacteria and archaea. The ISME Journal, 6(3), 610–618.

We also have written a paper in the area:

Matsen, F., & Gallagher, A. (2012). Reconciling taxonomy and phylogenetic inference: formalism and algorithms for describing discord and inferring taxonomic roots. Algorithms for Molecular Biology: AMB, 7(1), 8.

In it we address the ambiguity of intruder versus intruded (described above) is by casting it as a minimization problem, which is NP-complete but fixed-parameter tractable.

Experimental design

This paper is a software announcement rather than a research paper, so the “research question” doesn’t quite fit here.

The code is well commented, and the vignette does a nice job of showing functionality of the package. The reference manual seems complete. I could not find any sort of tests, which is unfortunate. I was surprised that the code does not handle multifurcating trees.

Validity of the findings

No Comments, as this is a software announcement.

Additional comments

My recommendation is somewhere between minor and major revisions. If it would seem helpful for me to look at the paper again, I'd be happy to do so, but I'm not going to insist.

Reviewer 3 ·

Basic reporting

The manuscript appears to meet all the standards for the journal.

Experimental design

The experimental design seems sound, apart from one small point. It would seem better not to report on the monophyly of taxa with one representative. The outcome is trivial, as there is no opportunity to not be monophyletic. Hence, restricting the evaluation to taxa with two more more representatives allows more meaningful results, and perhaps might save some computation. I do agree that reporting the number of taxa with a single representative is useful, but considering them as monophyletic when there is no possible way for them not to be monophyletic seems silly.

Validity of the findings

The findings seem valid.

Additional comments

Although my determination is accept as is, I think you should consider the point raised regarding evaluating and reporting monophyly for taxa with one representative.

---

## Round 0.2 · accepted · Accept

Thank you for making the changes requested by the reviewers.